# Validation of the Greek Cardiovascular Diet Questionnaire 2 (CDQ-2) and Single-Center Cross-Sectional Insights into the Dietary Habits of Cardiovascular Patients

**DOI:** 10.3390/nu17101649

**Published:** 2025-05-12

**Authors:** Konstantinos Giakoumidakis, Evridiki Patelarou, Anastasia A. Chatziefstratiou, Dimitra Aloizou, Nikolaos Vaitsis, Hero Brokalaki, Nikolaos V. Fotos, Elisabeth Geniataki, Athina E. Patelarou

**Affiliations:** 1Department of Nursing, School of Health Sciences, Hellenic Mediterranean University, 71410 Heraklion, Greece; epatelarou@hmu.gr (E.P.); ddk266@edu.hmu.gr (E.G.); apatelarou@hmu.gr (A.E.P.); 2Department of Nursing, School of Health Sciences, National and Kapodistrian University of Athens, 11527 Athens, Greece; a.chatziefstratiou@yahoo.gr (A.A.C.); heropan@nurs.uoa.gr (H.B.); nikfotos@nurs.uoa.gr (N.V.F.); 3ICU Department, IASO General and Maternity Hospital, 15123 Athens, Greece; dimitraloi@yahoo.gr; 4Primary Healthcare, 40300 Farsala, Greece; vaitsis@hotmail.com

**Keywords:** cardiovascular disease, cross-sectional study, diet, questionnaire, validation study

## Abstract

**Background/Objectives**: Dietary recommendations are an essential part of guidelines for the best management of chronic cardiovascular diseases. The present study aimed to validate the Greek version of Cardiovascular Diet Questionnaire 2 (CDQ-2) and to assess the dietary habits among cardiovascular patients. **Methods**: A single-center cross-sectional observational study was conducted. The study population was cardiovascular patients that were users of a private primary healthcare clinic. The data were collected between December 2024 and January 2025. The questionnaire was translated from French, back-translated, and reviewed by a committee of experts. The MEDAS was used as a gold standard. The psychometric measurements that were performed included reliability coefficients and Explanatory Factor Analysis (EFA). **Results**: The total sample comprised 90 individuals. The Cronbach’s α was 0.97. A bivariate Pearson’s correlation established that there was a strong, statistically significant linear relationship between the CDQ-2 and MEDAS scores, with r(90) = 0.962 and *p* < 0.001. Cardiovascular patients seemed to have suboptimal dietary patterns, as indicated by the relatively low mean CDQ-2 score of 2.9 (SD = 17.2), along with a mean MEDAS score of 8 (SD = 5.2), where younger individuals (*p* < 0.001), males (*p* = 0.042), single/divorced/widowed individuals (*p* < 0.001), individuals with lower physical activity (*p* = 0.001), and active smokers (*p* = 0.022) demonstrated significantly poorer adherence to the optimal cardiovascular dietary status. **Conclusions**: The survey indicated that CDQ-2 was a valid and reliable scale to use in cardiovascular patients in Greece. Also, the patients were characterized by suboptimal dietary habits, indicating the need for personalized interventions to improve their dietary habits.

## 1. Introduction

Cardiovascular disease (CVD) is the foremost cause of death all over the world, contributing to approximately 18 million deaths annually [1]. In Greece, CVD continues to be the leading cause of mortality and morbidity, despite the low rates recorded in the 1950s and 1960s, when the country was considered privileged in terms of cardiovascular health. According to the ATTICA epidemiological study, the incidence of cardiovascular disease in Greece reached 360 cases per 10,000 individuals over a 20-year follow-up period (2002–2022), indicating a high lifetime risk of developing cardiovascular disease [2]. Beyond the estimation of cardiovascular disease-related deaths, there is growing concern about its substantial burden, as it ranks first compared with other diseases. In 2021, CVD was responsible for 14.9% of the total loss of healthy life years due to either premature mortality or disability [3,4].

Dietary modifications play a pivotal role in the secondary and tertiary prevention of cardiovascular disease. According to the literature, a diet rich in unsaturated fats and low in saturated fats is highly beneficial for patients with cardiovascular diseases, including those who have undergone Coronary Artery Bypass Grafting (CABG), as well as those with other forms of CVD, such as heart failure, hypertension, and ischemic heart disease. Many studies have documented the value of a well-balanced diet in beneficial patient outcomes, including healthier metabolic profiles, reduced inflammation, enhanced overall cardiovascular function, and diminished mortality rates [5,6].

For instance, a diet rich in fruits, vegetables, whole-grain cereals, and lean proteins has been consistently associated with better clinical outcomes and fewer complications in patients with cardiovascular diseases. These dietary patterns help regulate blood pressure, improve lipid profiles, and reduce the risk of recurrent cardiovascular events [7]. The Mediterranean diet is widely considered to be the most appropriate dietary model for cardiovascular patients due to its emphasis on plant-based foods, healthy fats (such as olive oil), and the moderate consumption of fish and poultry. This diet has been shown to significantly reduce mortality; morbidity; and complications, such as angina and acute myocardial infarction [8,9].

Beyond its physical health benefits, the Mediterranean diet also positively impacts psychological well-being. Unsanitary dietary habits, such as the high consumption of processed foods and saturated fats, have been linked to increased levels of anxiety, stress, and depression, which can deteriorate cardiovascular outcomes [10,11]. For instance, patients with heart failure who adhere to a poor diet pattern often experience higher levels of psychological distress, which can worsen their condition, lessening their quality of life [12]. Additionally, cognitive impairments, such as delirium and mild cognitive decline, are more frequent among cardiovascular patients with improper dietary habits, further underscoring the importance of a healthy diet in maintaining both physical and mental health [12].

Despite the well-documented benefits of a healthy diet, adherence to dietary recommendations remains suboptimal for individuals with various forms of cardiovascular disease. Studies have identified several barriers to dietary adherence, such as a lack of patient education, limited access to healthy foods, socioeconomic challenges, cultural factors, and insufficient follow-up care [13,14]. The first and crucial step to confront this issue is the reliable estimation of the nutritional status of these patients to plan and perform specific and systematic interventions to try to reverse the existing problematic situation and alleviate all the associated obstacles that impede the adoption of a healthy diet pattern among cardiovascular patients.

Cardiovascular Diet Questionnaire 2 (CDQ-2) is a validated dietary assessment tool originally developed for the French population by Paillard et al. [15] and designed to evaluate dietary habits in patients with cardiovascular diseases. According to its nature and content, it provides a comprehensive assessment of nutrient intake based on a seven-day dietary history and biomarkers. Additionally, this tool captures both qualitative and quantitative aspects of a diet, making it a robust tool for clinical and research settings [15].

The purpose of the present study was (a) to translate CDQ-2 into the Greek language and validate it for the Greek-speaking population of cardiovascular patients and (b) to assess their dietary status, identifying the factors influencing it. This study intended to add useful new data to the existing body of literature by providing a culturally adapted and validated dietary tool for the Greek-speaking population, which can serve many research and clinical purposes in the field of cardiovascular prevention and holistic care. Additionally, it could provide a comprehensive understanding of the current dietary status of cardiovascular patients and the influencing factors, offering insights into tailoring personalized interventions and policies.

## 2. Materials and Methods

### 2.1. Study Design

This was a single-center cross-sectional observational study. The study population was cardiovascular disease patients that were users of a private primary healthcare clinic. The convenience sampling method was used to collect data from December 2024 to January 2025. Participants who were eligible to take part in this study needed to be men or women aged 18 years and above who had been diagnosed with cardiovascular disease. Moreover, they needed to be able to read and write in Greek to ensure they could understand the study materials and take part in the related assessments effectively. Only individuals who willingly agreed to participate in this study were included in the enrollment process. As exclusion criteria, we used a history of psychiatric illness, a recent history of alcohol and/or drug abuse, dementia, and Alzheimer’s disease (Figure 1).

### 2.2. Translation of CDQ-2

The process involved an independent translation of the original French as a forward translation by two separate individuals. After this phase, a third individual compared the two translations and was able to determine an agreed-upon translation (1st reconciliation version). A bilingual individual, whose mother tongue was French and was a professional translator, later translated the agreed-upon version into the original questionnaire’s language (backward translation). However, the individual was unaware of the questionnaire’s standard format.

Twenty patients with cardiovascular disease completed the translated version of the questionnaire to evaluate its apparent validity (face validity), which verified that the scale included questions that were relevant to the characteristic being measured and did not lead to inaccurate or incomplete responses. Additionally, three eHealth specialists reviewed the questionnaire as part of a cognitive debriefing to assess the tool and suggest ways to make it better (Content Validity). In the final version of the instrument (Appendix A), items with a Coefficient Validity Ratio (CVR) greater than 0.70 were retained.

### 2.3. Reliability

To measure a scale’s reliability, the internal consistency coefficient, Cronbach’s alpha, was calculated. This coefficient evaluates the degree to which the questions that make up a scale measure the same concept. Values greater than or close to 0.70 (70%) are acceptable. An internal consistency coefficient between 0.50 and 0.60 (50–60%) is considered sufficient in the early stages of a study. If the value exceeds 0.80 (80%), then it is considered a particularly good reliability analysis.

### 2.4. Validity

Confirmatory factor analysis (CFA) was conducted to determine the model’s fitness. An adequate or good fit was indicated by a Standardized Root-Mean-Squared Residual (SRMR) less than or equal to 0.08, a Coefficient of Determination (CD) greater than or equal to 0.90, and a Comparative Fit Index (CFI) greater than or equal to 0.90.

The construct validity was determined using Pearson’s correlation between CDQ-2 and the selected gold standard questionnaire.

### 2.5. Translation and Weighting Questionnaire

Data were collected using an anonymous self-report questionnaire that consisted of three sub-sections:-The first concerned the following demographic characteristics: (a) age, (b) biological sex, (c) educational level, (d) subjective economic situation, and (e) family situation.-CDQ-2 includes 17 closed-ended questions designed to identify major sources of nutrients. For each of the 9 questions related to the saturated fatty acid (SFA) intake, the total score ranges from 0 to 27. The monounsaturated fatty acid (MUFA) intake is investigated with one question (total score range 0–6). Omega-3 fatty acid (ω3FA) intake is investigated with 3 questions (total score range 0–10). Finally, there are 4 questions on fruit and vegetable (FV) consumption (range 0–14). The total nutritional score is calculated as [(FV + MUFA + ω3FA) − SFA], fluctuating from −27 to +30. Higher scores indicate better nutrition, whereas the existing literature does not provide universal cut-off points.

MEDAS was created by Spanish researchers [16] and has been weighted and translated by Greek researchers [17]. This tool consists of 14 questions (total score range 0–14) about the main food groups consumed as part of the Mediterranean Diet, which is a valid means of rapidly assessing adherence to it. Higher scores indicate the healthiest dietary pattern, while a score ≥ 10 is considered indicative of high adherence [18]. It is known that the dietary pattern associated with the Mediterranean Diet is characterized by the daily consumption of olive oil (mainly extra virgin), whole grains, fruits, and vegetables. The Mediterranean diet is a rich source of essential minerals, vitamins, and fiber and is considered one of the healthiest dietary patterns.

### 2.6. Statistical Methodology

We performed the statistical analysis using STATA software (version 12.0; Stata Corporation, College Station, TX, USA) for the confirmatory factor analysis and SPSS statistical software (version 27; SPSS, Chicago, IL, USA) for the remainder. For the descriptive statistical analysis, the continuous variables are given as the mean value and standard deviation, while the discrete ones are given as the absolute and relative frequencies. Multiple Logistic Regression was used to investigate the independent variables (age, biological sex, higher level of education, economic level, marital status, physical activity, smoking, and alcohol consumption) that can predict the dependent variable (CDQ-2). The minimum value of the statistical significance level was set at 5%. To weigh the CDQ-2 questionnaire, the following steps were followed: (a) bilingual translation (forward translation, reconciliation report, backward translation), (b) cognitive debriefing process of the questionnaire through the pilot data collection of a small sample of participants (10–15 people) to fully oversee the formation of the final form of the translated questionnaire, (c) calculation of the questionnaire reliability using the repeatability method, (d) calculation of the concurrent validity (the MEDAS questionnaire was used as the gold standard), and (e) calculation of validity through confirmatory factor analysis. For the confirmatory factor analysis, the calculation of the minimum required sample size showed that for 17 items, a single-factor model, a type I error (α) of 5%, a statistical power of 80%, and an expected effect size of 0.1, at least 87 participants were required. This requirement was met, as the total sample size enrolled in the present study was 90 individuals.

## 3. Results

The total sample comprised 90 individuals, where 13 (14.4%) were female and 77 (85.6%) were male; the mean age was 63.8 ± 9.6 years. A total of 30 (33.3%) participants were single, divorced, or widowed, and 60 (66.7%) were married or cohabited. Regarding the education level, 21 (23.3%), 41 (45.6%), 21 (23.3%), and 7 (7.8%) participants had up to secondary education, post-secondary non-tertiary education, tertiary education, and held an MSc or PhD, respectively. A total of 24 (26.7%) participants reported a low economic level, while 50 (55.6%) and 16 (17.8%) reported middle and high economic levels, respectively. In terms of physical activity, 32 (35.6%) participants had a low level, 43 (47.8%) a middle level, and 15 (16.7%) a high level. Regarding alcohol consumption, 19 (21.1%) reported no consumption of alcohol, 55 (61.1%) moderate consumption, and 16 (17.8%) high consumption (Table 1). The CVR results for CDQ-2 showed that 100% of the items (n = 17) were acceptable, and the Cronbach’s α was 0.97. A one-factor model conducted by CFA yielded acceptable global fit indices (SRMR = 0.08, CD = 0.99, CFI = 0.99), indicating that the 17 items in the one-factor solution proposed by the principal investigators should be accepted for the Greek version of CDQ-2 (Figure 2). The mean score of CDQ-2 was 2.9 (SD = 17.2), and the mean MEDAS score was 8 (SD = 5.2). A bivariate Pearson’s correlation established that there was a strong, statistically significant linear relationship between the CDQ-2 and MEDAS scores, with r (90) = 0.962 and *p* < 0.001.

A multivariate analysis was performed to ascertain the effects of the independent variables, namely, age, biological sex, level of education, economic level, marital status, physical activity, smoking, and alcohol consumption, on the CDQ-2 score. The multiple linear regression model was statistically significant, with F (8, 81) = 75.4 and *p* < 0.001. The model explained 87% (Adjusted R-Square) of the variance in the score of CDQ-2. As shown in Table 2, of the eight predictor variables, only the following five were statistically significant: (1) age (*p* < 0.001), (2) biological sex (*p* = 0.042), (3) marital status (*p* < 0.001), (4) physical activity (*p* = 0.001), and (5) smoking (*p* = 0.022). Specifically, younger age, male sex, being single/divorced/widowed, lower physical activity levels, and active smoking status were strongly associated with lower CDQ-2 scores (Table 2). Table 3 displays the distribution of CDQ-2 scores according to the categories of the categorical variables.

## 4. Discussion

The present study contributed two main points: first, it documented the suboptimal adherence to healthy dietary patterns among patients with cardiovascular disease, as demonstrated in the low mean scores of both the MEDAS and CDQ-2 tools, and second, it validated the Greek version of CDQ-2 as a culturally adapted, valid, and reliable instrument for assessing the diet quality in this patient population. Moreover, the multivariate analysis revealed several socio-demographic variables, namely, age, sex, marital status, physical activity levels, and smoking status, that were substantial predictors of lower CDQ-2 scores and poor dietary adherence.

Specifically, the participants were not characterized by optimal adherence to proper diet habits. Despite the benefits of optimal diet habits for patients with cardiovascular diseases being distinct and well-documented in terms of lower morbidity [19,20] and mortality rates [20,21], many studies were in line with our findings, revealing inadequate adherence observed, even to the Mediterranean Diet among the inhabitants of Mediterranean countries [22,23,24]. This finding is indicative of poor self-management and self-care behavior, jeopardizing the secondary and tertiary prevention of cardiovascular disease. Adopting suboptimal adherence to a well-known advantageous diet archetype constitutes a global issue that may be attributed to systemic problems in patient education, nutritional counseling, low healthcare literacy, and limited access to structured and well-organized supportive services.

The low mean scores observed in the MEDAS (8, SD = 5.2) and CDQ-2 (2.9, SD = 17.2), even in a population thought to be highly motivated to change their lifestyle, have important clinical implications, underscoring the clinical importance of the early identification of patients with inadequate dietary habits and the need for tailored nutritional interventions to improve long-term cardiovascular outcomes [8,20]. Furthermore, these findings highlight the urgent need for more systematic, personalized, and non-generic nutritional interventions targeting high-risk patient groups [9,19].

Additionally, younger patients exhibited lower CDQ-2 scores, suggesting diminished adherence to the recommended eating patterns. This finding is in line with prior studies indicating that older individuals are generally more compliant with healthy dietary recommendations and guidelines, potentially due to increased awareness of secondary prevention measures and existing comorbidities [1,2]. Younger patients, even those already diagnosed with cardiovascular disease, may underestimate their risk, leading to a false sense of security, which impedes them from activating healthier diet habits. In contrast, older patients are more likely to adhere to healthier dietary patterns, possibly motivated by the presence of multiple health disorders and concerns about severe cardiovascular events.

Similarly, male sex was found to be significantly associated with lower CDQ-2 scores, suggesting a gender gap in dietary adherence among cardiovascular patients. This is consistent with findings from previous studies indicating that women are more likely to engage in health-promoting behaviors, including healthier eating habits and greater compliance with dietary recommendations [25,26]. Possible explanations for this disparity include higher health literacy levels among women, increased involvement in food preparation, and greater concern for long-term health outcomes. Additionally, traditional gender roles and sociocultural perceptions around masculinity may discourage men from prioritizing dietary changes, especially in older age groups [27].

Regarding marital status, being single, divorced, or widowed was associated with significantly lower adherence to cardiovascular dietary guidelines. Several studies have highlighted the protective effect of marriage or cohabitation on health behaviors and outcomes, including diet quality [28,29]. Living with a partner may provide emotional support, shared responsibility in meal planning, and accountability, which contribute to better adherence. Conversely, individuals who live alone may experience lower motivation to maintain structured eating habits or may resort to convenience and processed foods, especially in the context of older age and chronic illness [30].

The physical activity level was another significant determinant of CDQ-2 scores, with lower activity levels correlating with poorer dietary patterns. This finding aligns with literature suggesting that health-related behaviors tend to cluster together, with physically active individuals more likely to engage in healthy eating [31]. It has been proposed that physical activity improves self-regulation and health consciousness, which may influence dietary decisions. Additionally, individuals who exercise regularly are often exposed to health promotion environments or programs that reinforce positive dietary habits [32].

Active smoking was independently associated with lower CDQ-2 scores. Smokers are generally less likely to follow health guidelines, possibly due to a lower perceived vulnerability to disease or a higher tendency toward risk-taking behavior [33]. Moreover, smoking often co-occurs with other unhealthy lifestyle choices, including poor dietary habits, low physical activity, and high alcohol consumption [34]. This behavioral clustering underlines the importance of integrated lifestyle interventions that address multiple modifiable risk factors simultaneously in cardiovascular patients.

Additionally, this was a clinical study aimed at translating and validating a disease-specific dietary tool, namely, CDQ-2, for assessing dietary habits among patients with cardiovascular disease, based on a seven-day dietary history and biomarkers. The Greek version of CDQ-2 demonstrated good construct and face validity. The Cronbach’s alpha was 0.97 for the entire scale, which confirmed the internal consistency of the tool in line with the validation analysis. According to the CFA, a one-factor solution that comprised all 17 items was proposed and accepted for the Greek version of CDQ-2. Therefore, CDQ-2 is a valid and reliable tool that is recommended for use as an overall scale. Moreover, CFA indicated acceptable global fit indices (SRMR = 0.08, CD = 0.99, CFI = 0.99).

We used the MEDAS questionnaire as the gold standard to calculate the concurrent validity of CDQ-2. The MEDAS assessed the main food groups consumed in the Mediterranean Diet. The analysis showed that the mean CDQ-2 total score was 2.9 (SD = 17.2), while the mean MEDAS total score was 8 (SD = 5.2). A bivariate Pearson’s correlation revealed a strong, statistically significant linear relationship between the CDQ-2 and MEDAS total scores, with r (90) = 0.962 and *p* < 0.001. This finding underscores the robustness of CDQ-2 in estimating dietary adherence within the cardiovascular setting.

### Study’s Strengths and Limitations

Beyond the strengths outlined above and its contribution of new knowledge to the current body of literature by offering a culturally adapted and validated tool for the dietary assessment of Greek-speaking patients suffering from cardiovascular disorders, the present study had some limitations. First, the cross-sectional design and the single-center nature pose significant threats to the generalizability of the findings to a broader cardiovascular population, as well as to the establishment of causal relationships between dietary habits and patient characteristics or outcomes. Second, the convenience sampling method, combined with the existence of a gender-unbalanced sample, with a greater proportion of males, raises concerns regarding the generalization of the dietary assessment and adherence findings. Last but not least, the potential overestimation or underestimation of participants’ dietary habits due to self-report bias may represent an additional issue of the present study.

## 5. Conclusions

This survey supported the validity and reliability of CDQ-2 among cardiovascular patients in Greece, while the strong correlation between the CDQ-2 and MEDAS scores underscored the robustness of CDQ-2 in assessing dietary adherence within this population group. The cardiovascular patients seemed to have suboptimal dietary patterns, as indicated by the relatively low mean CDQ-2 score of 2.9 (SD = 17.2), along with a mean MEDAS score of 8 (SD = 5.2), with younger individuals, males, single/divorced/widowed individuals, individuals with lower physical activity, and active smokers demonstrating poorer adherence to the optimal and beneficial cardiovascular dietary status. Healthcare providers could incorporate CDQ-2 in routine clinical practice to identify patients with a lower capacity to follow and adopt the recommended dietary modifications. Future research is needed to inform clinical practitioners and support the design of tailored and personalized interventions for improving dietary behaviors among cardiovascular patients. Also, future validation efforts should aim to include more gender-balanced and representative samples from multiple centers to confirm the applicability of CDQ-2, which would be more informative in terms of the generalization and exportation of the causal associations between variables.

## Figures and Tables

**Figure 1 nutrients-17-01649-f001:**
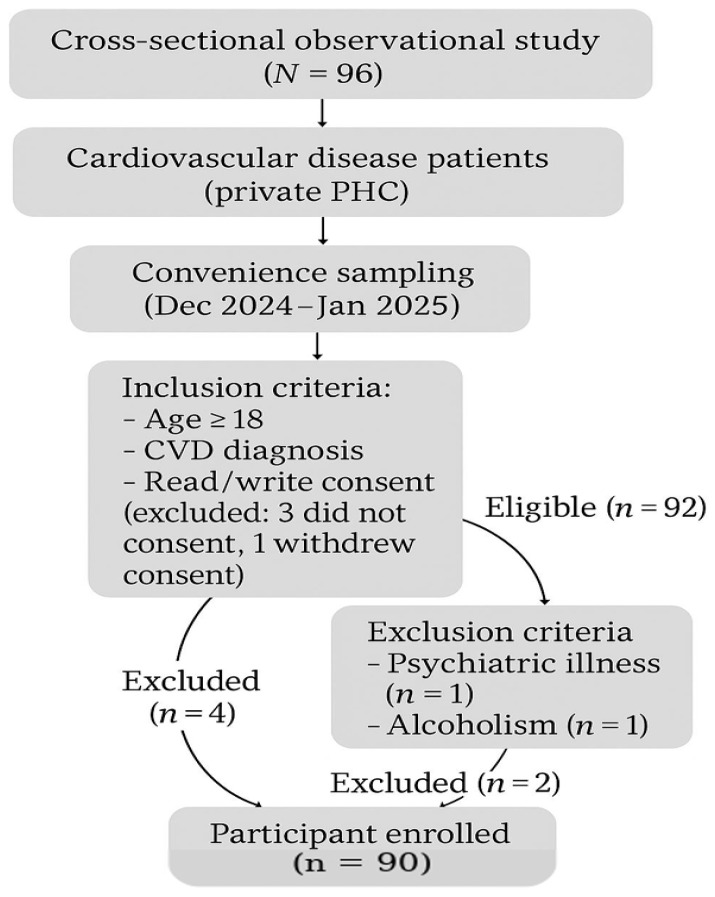
Participant flowchart. CVD: cardiovascular disease, PHC: primary healthcare.

**Figure 2 nutrients-17-01649-f002:**
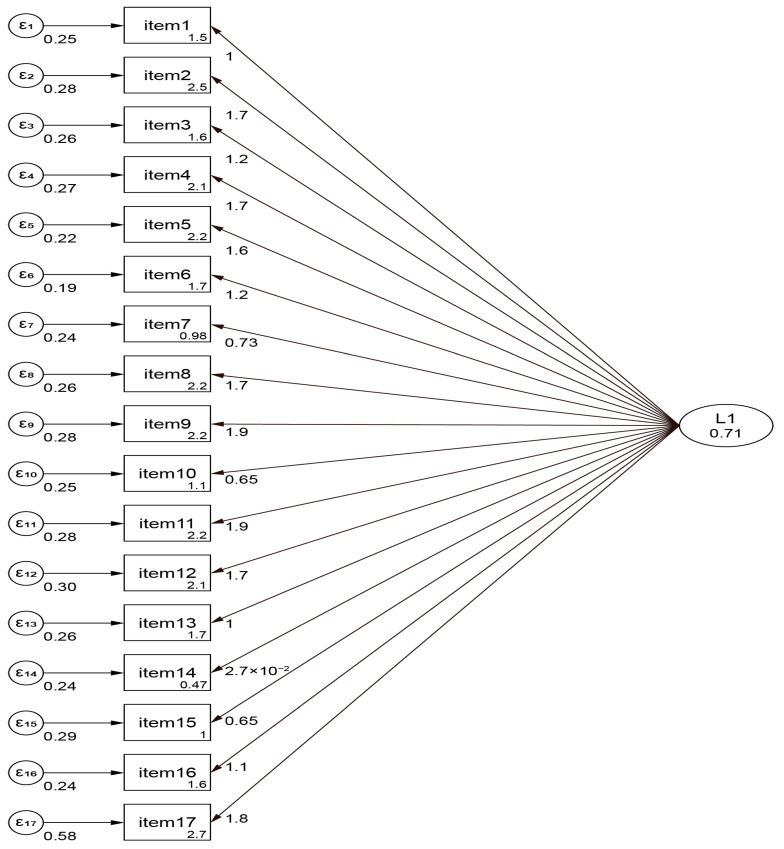
Confirmatory factor analysis of CDQ-2.

**Table 1 nutrients-17-01649-t001:** Sample characteristics.

		N	N %
Age (years)		63.8 ± 9.6 *	
Biological sex	Female	13	14.4%
	Male	77	85.6%
Higher level of education	Up to secondary education	21	23.3%
	Post-secondary non-tertiary education	41	45.6%
	Tertiary education	21	23.3%
	MSc/PhD	7	7.8%
Economic level	Low	24	26.7%
	Middle	48	53.3%
	High	18	20.0%
Marital status	Single, divorced, or widowed	30	33.3%
	Married or cohabiting	60	66.7%
Physical activity	Low	32	35.6%
	Middle	43	47.8%
	High	15	16.7%
Smoking	Smoker	17	18.9%
	Ex-smoker	54	60.0%
	Non-smoker	19	21.1%
Alcohol consumption	Medium consumption	19	21.1%
	Social consumption	55	61.1%
	No consumption	16	17.8%

(*) mean ± standard deviation.

**Table 2 nutrients-17-01649-t002:** Contribution of each independent variable to the model and its statistical significance.

	Unstandardized Coefficients		Standardized Coefficients	t	Sig.	95.0% Confidence Interval for B	
	B	Std. Error	Beta			Lower Bound	Upper Bound
(Constant)	63.665	20.479		3.109	0.003	22.918	104.412
Age	−1.181	0.234	−0.660	−5.052	0.000	−1.647	−0.716
Biological sex	5.041	2.438	0.104	2.068	0.042	0.191	9.891
Higher level of education	−0.564	1.647	−0.029	−0.342	0.733	−3.842	2.714
Economic level	1.161	2.202	0.046	0.527	0.600	−3.221	5.543
Marital status	10.905	2.118	0.301	5.149	0.000	6.691	15.120
Physical activity	4.500	1.355	0.184	3.321	0.001	1.804	7.196
Smoking	−6.399	2.733	−0.237	−2.341	0.022	−11.837	−0.960
Alcohol	3.473	2.496	0.127	1.391	0.168	−1.493	8.439

Dependent variable: CDQ-2.

**Table 3 nutrients-17-01649-t003:** The mean score of CDQ-2 for a studied factor.

		CDQ-2	
		Mean	Standard Deviation
Biological sex	Female	19.3	5.4
	Male	0.1	16.9
Higher level of education	Up to secondary education	−19.1	1.8
	Post-secondary non-tertiary education	2.5	13.9
	Tertiary education	19.6	1.2
	MSc/PhD	21.1	1.7
Economic level	Low	−15.6	10.3
	Middle	6.9	13.6
	High	16.8	12.5
Marital status	Single, divorced, or widowed	−16.7	4.8
	Married or cohabiting	12.7	11.8
Physical activity	Low	−14.2	7.3
	Middle	9.7	14.5
	High	19.9	1.4
Smoking	Smoker	−14.0	12.5
	Ex-smoker	3.9	15.2
	Non-smoker	15.1	14.3
Alcohol consumption	Medium consumption	−13.6	11.8
	Social consumption	3.3	15.7
	No consumption	21.1	0.9

## Data Availability

The data that support the findings of this study are available from the corresponding author upon reasonable request due to privacy restrictions.

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
