# Peer review of "Validation of the Greek Cardiovascular Diet Questionnaire 2 (CDQ-2) and Single-Center Cross-Sectional Insights into the Dietary Habits of Cardiovascular Patients"

_nutrients, 2025, doi:10.3390/nu17101649_

Round 1
Reviewer 1 Report
Comments and Suggestions for Authors
The manuscript addresses a relevant aspect of dietary assessment in cardiovascular patients, namely the validation of an adapted tool (the Cardiovascular Diet Questionnaire 2) for use in the Greek patients. Nevertheless, the current version of the manuscript has several methodological limitations that require major revision. Please find below my comments and suggestions:
1. The title suggests a substantial focus on the dietary assessment of cardiovascular patients; however, the results relevant to this objective are presented in a limited and insufficient manner. The dietary assessment of the study population is only partially developed, and lacks both statistical rigour and clinical interpretation.
2. The Material and Methods, particularly the selection of participants, raise concerns. The sample was predominantly male participants (approximately 85%), which introduces significant gender bias and limits the generalisability of the findings.
3. The Results section lacks comprehensiveness. In addition, the assessment of validity is only partially addressed through correlation with the MEDAS. A more robust validation should include both validity (comparison with MEDAS results) and reproducibility (comparison of results obtained twice - the Greek version of the CDQ-2). The lack of a Bland-Altman plot and corresponding indices limits the completeness of the assessment.
4. Moreover, the low mean CDQ-2 and MEDAS scores are reported without adequate discussion of the clinical implications of these results.
5. The discussion section is overly and lacks a coherent synthesis of the main contributions of the study. Importantly, there is no separate subsection discussing the strengths and limitations of the study. Furthermore, the methodological elements described around lines 294-297 would be more appropriately placed in the Materials and Methods section and should be revised accordingly.
6. The full version of the CDQ-2 questionnaire should be included as supplementary material. This would increase the transparency of the validation process and facilitate its use by other researchers and clinicians in similar contexts.
Author Response
Dear Reviewer 1,
Thank you very much for your valuable comments, which have significantly contributed to improving our paper. Please find below our point-by-point responses to your suggestions. We believe that all the issues you raised have now been fully addressed, and the revisions meet your expectations.
On behalf of all authors,
Dr. Konstantinos Giakoumidakis
Associate Professor, Department of Nursing
School of Health Sciences, Hellenic Mediterranean University, Heraklion, Crete, Greece

Reviewer 2 Report
Comments and Suggestions for Authors
The manuscript titled “nutrients-3620856_Dietary Assessment in Cardiovascular Patients: Cross-Sectional Insights and Validation of the Greek Cardiovascular Diet Questionnaire 2 (CDQ-2)” has been submitted to the “Clinical Nutrition” section of the Special Issue “Nutrition and Quality of Life for Patients with Chronic Disease”.
The aim of this study was to validate the Greek version of the Cardiovascular Diet Questionnaire 2 (CDQ-2) and to assess dietary habits among patients with cardiovascular disease.
Comments:
The subject matter of the submitted manuscript aligns well with the focus of the section and the Special Issue to which it was submitted.
The title is informative and accurately reflects the content of the manuscript.
The abstract is well structured and includes the key information relevant to the study.
The keywords are appropriate and consistent with the content of the paper.
Introduction: The introduction effectively highlights the importance of cardiovascular diseases, with a particular focus on the situation in Greece. It also discusses the newly developed questionnaire in France and its potential applicability to the Greek population, supported by appropriate references.
Materials and Methods: The section does not mention approval from an ethics committee, which is essential for the implementation of this type of study. Please ensure that ethical approval is explicitly stated.
Results: The results are clearly presented; however, further elaboration and explanation would improve their interpretation. Figure 1 is difficult to read and should be revised for better clarity.
Discussion: The discussion provides a sound interpretation of the findings within the context of existing literature, using relevant references. However, the limitations of the study—such as the sample size and gender distribution—should be addressed more explicitly.
Conclusion: The conclusion is consistent with the study's objective and appropriately summarizes the findings.
Author Response
Dear Reviewer 2,
Thank you very much for your valuable comments, which have significantly improved our paper. Please find below our point-by-point responses to your suggestions. We believe that all the issues you raised have now been fully addressed, and the revisions meet your expectations.
On behalf of all authors,
Dr. Konstantinos Giakoumidakis
Associate Professor, Department of Nursing
School of Health Sciences, Hellenic Mediterranean University, Heraklion, Crete, Greece

Round 2
Reviewer 1 Report
Comments and Suggestions for Authors
The authors have addressed the comments raised during the first round of peer review. However, several aspects of the manuscript require refinement and elaboration, as outlined below:
1. The title and abstract broadly reflect the focus of the study. However, as the research was conducted at a single centre, it is recommended that the title be revised to explicitly state this, e.g. A single-centre cross-sectional study.
2. In the Materials and Methods section, the inclusion of a flowchart showing patient enrolment and detailed inclusion and exclusion criteria is strongly recommended to improve methodological transparency.
3. The single-centre nature of the study should be clearly stated and consistently emphasised throughout the manuscript, particularly in the Material and Methods section.
4. In the Results section, a more thorough discussion of the implications and originality of the findings is encouraged to better highlight the contribution of the study to the existing body of knowledge.
5. Finally, the Discussion section should conclude with a separate section addressing the strengths and limitations of the study.
Author Response
Dear Reviewer 1,
Once again, thank you very much for your valuable comments, which have significantly contributed to improving our paper. Please find below our point-by-point responses to your suggestions. We believe that all the issues you raised have now been fully addressed, and the revisions meet your expectations.
On behalf of all authors,
Dr. Konstantinos Giakoumidakis
Associate Professor, Department of Nursing
School of Health Sciences, Hellenic Mediterranean University, Heraklion, Crete, Greece
Reviewer #1 (round 2):
The authors have addressed the comments raised during the first round of peer review. However, several aspects of the manuscript require refinement and elaboration, as outlined below:
- The title and abstract broadly reflect the focus of the study. However, as the research was conducted at a single centre, it is recommended that the title be revised to explicitly state this, e.g. A single-centre cross-sectional study.
Response: Many thanks for your valuable comment. We have now revised the title to “Validation of the Greek Cardiovascular Diet Questionnaire 2 (CDQ-2) and Single-Centre Cross-Sectional Insights into the Dietary Habits of Cardiovascular Patients” based on your kind recommendation.
- In the Materials and Methods section, the inclusion of a flowchart showing patient enrolment and detailed inclusion and exclusion criteria is strongly recommended to improve methodological transparency.
Response: Thank you for this valuable suggestion. As recommended, we have included a flowchart named “Participant flowchart” (Figure 1, line 114) illustrating patient enrolment and the inclusion and exclusion criteria to enhance methodological transparency. Also, we renamed the Figure 1 to Figure 2 (line 235).
- The single-centre nature of the study should be clearly stated and consistently emphasised throughout the manuscript, particularly in the Material and Methods section.
Response: Many thanks for your valuable comment. We have now incorporated clear statements about the single-centre nature of the present study in lines 3, 19, 102, and 325, maintaining at the same time a balanced approach to avoid unnecessary repetition.
- In the Results section, a more thorough discussion of the implications and originality of the findings is encouraged to better highlight the contribution of the study to the existing body of knowledge.
Response: Many thanks for your comment. When writing a paper, we always adhere to the established guidelines for each manuscript section. The Results section provides a complete description and reference of the study’s findings, while no interpretation or discussion should be included. This approach is well-documented in literature; for instance, you may refer to the following:
- Forero DA, Lopez-Leon S, Perry G. A brief guide to the science and art of writing manuscripts in biomedicine. J Transl Med. 2020 Nov 10;18(1):425. doi: 10.1186/s12967-020-02596-2. PMID: 33167977; PMCID: PMC7653709.
- Busse C, August E. How to Write and Publish a Research Paper for a Peer-Reviewed Journal. J Cancer Educ. 2021 Oct;36(5):909-913. doi: 10.1007/s13187-020-01751-z. PMID: 32356250; PMCID: PMC8520870.
- Gemayel R. How to write a scientific paper. FEBS J. 2016 Nov;283(21):3882-3885. doi: 10.1111/febs.13918. PMID: 27870269.
We fully agree with your suggestion to highlight the contribution of our study to the existing body of literature. For this reason, we have emphasized it in several parts of the paper, for example in lines 238–245, 305–313, 322–324, and 335–344.
- Finally, the Discussion section should conclude with a separate section addressing the strengths and limitations of the study.
Response: We truly appreciate your valuable comment. According to your suggestion, we have now added a separate subsection addressing the strengths and limitations of the study (lines 321-333).